# Spatially Modulated Fiber Speckle for High-Sensitivity Refractive Index Sensing

**DOI:** 10.3390/s23156814

**Published:** 2023-07-31

**Authors:** Penglai Guo, Huanhuan Liu, Zhitai Zhou, Jie Hu, Yuntian Wang, Xiaoling Peng, Xun Yuan, Yiqing Shu, Yingfang Zhang, Hong Dang, Guizhen Xu, Aoyan Zhang, Chenlong Xue, Jiaqi Hu, Liyang Shao, Jinna Chen, Jianqing Li, Perry Ping Shum

**Affiliations:** 1School of Computer Science and Engineering, Faculty of Innovation Engineering, Macau University of Science and Technology, Avenida Wai Long, Taipa, Macau 999078, China; 2009853yii30003@student.must.edu.mo (P.G.);; 2Department of Electronic and Electrical Engineering, College of Engineering, Southern University of Science and Technology, 1088 Xueyuan Avenue, Shenzhen 518055, China; liuhh@sustech.edu.cn (H.L.); shum@ieee.org (P.P.S.); 3State Key Laboratory of Optical Fiber and Cable Manufacture Technology, Southern University of Science and Technology, Shenzhen 518055, China; 4Key Laboratory of Optoelectronics Integrated Circuit Intellisense of Guangdong, Southern University of Science and Technology, Shenzhen 518055, China; 5Guangdong-Hong Kong-Macao Joint Laboratory for Intelligent Micro-Nano Optoelectronic Technology, Foshan University, 18 Jiang-Wan-Yi-Lu, Foshan 528000, China; 6Pengcheng Laboratory, 2 Xingke 1st Street, Shenzhen 518055, China

**Keywords:** optical fiber sensor, refractive index sensing, speckle, tapered multimode fiber

## Abstract

A fiber speckle sensor (FSS) based on a tapered multimode fiber (TMMF) has been developed to measure liquid analyte refractive index (RI) in this work. By the lateral and axial offset of input light into TMMF, several high-order modes are excited in TMMF, and the speckle pattern is spatially modulated, which affects an asymmetrical speckle pattern with a random intensity distribution at the output of TMMF. When the TMMF is immersed in the liquid analyte with RI variation, it influences the guided modes, as well as the mode interference, in TMMF. A digital image correlations method with zero-mean normalized cross-correlation coefficient is explored to digitize the speckle image differences, analyzing the RI variation. It is found that the lateral- and axial-offsets-induced speckle sensor can enhance the RI sensitivity from 6.41 to 19.52 RIU^−1^ compared to the one without offset. The developed TMMF speckle sensor shows an RI resolution of 5.84 × 10^−5^ over a linear response range of 1.3164 to 1.3588 at 1550 nm. The experimental results indicate the FSS provides a simple, efficient, and economic approach to RI sensing, which exhibits an enormous potential in the image-based ocean-sensing application.

## 1. Introduction

In recent years, fiber optical sensors have attracted much attention in marine applications because of their merits, including passive devices, remote sensing ability, simple structure, small size, low loss, strong anti-electromagnetic interference capability, etc. [1,2,3]. The measurement of seawater salinity is of especially great practical value in fields such as oceanography research and ocean environment monitoring [4,5]. Compared with a conventional conductivity–temperature–depth instrument, optical fiber sensors have demonstrated high sensitivity and good accuracy in the measurement refractive index (RI) of liquid analyte [6]. The RI of seawater is closely related to its salinity; the measurement of seawater salinity can be achieved by evaluating the RI of seawater. So, optical fiber sensors are expected to provide innovative solutions for seawater salinity measurement. Common types of RI fiber sensors include fiber Bragg grating sensors (FBG) [7], long-period grating sensors (LPG) [8], surface plasmon resonance sensors (SPR) [9], cladding mode resonance [10], and sensors based on the Fabry–Perot (FP) interferometer [11], Mach-Zehnder interferometer (MZI) [12], Michelson (ML) interferometer [13], etc. Typically, all these sensors are based on the resonance-wavelength shift principle [14], which requires broadband lasers and optical spectrum analyzers [15,16,17]. However, an optical spectral analyzer with high resolution is expensive, and the measurement time is determined by the wavelength scanning speed, limiting the wavelength-modulated fiber RI sensor in the application scenario where low cost and fast detection are required. As an alternative solution, intensity-modulated fiber RI sensors have been developed where the optical power is measured [18]. However, the power deviations from a light source affects the measurement accuracy of the sensors, resulting in the need to consider power compensation techniques [19].

To overcome the above limitation, the fiber speckle sensor (FSS) has become of great interest nowadays [19]. On the one hand, it replaces the expensive optical spectrum analyzer with an intensity-based camera with fast response. On the other hand, a new free degree in spatial intensity distribution is explored to achieve high sensitivity by the pixel array. It has been successfully applied to displacement [20], pressure [21], magnetic field [22], wavelength [23], DNA hybridization [24], and even the vector parameter [25]. To obtain the speckle pattern, standard single-mode fiber (SMF) should be replaced by special fiber supporting multiple modes. The modulation of these modes by the environment RI is the basis for RI measurement using a speckle sensor [26]. Currently, E. Fujiwara et al. proposed a concatenated structure and a special multimode fiber (MMF) by using no-core fiber and exposed-core MMF as the sensing heads, respectively [27,28]. The sensitivity of both sensors has been characterized to be 18.7 and 10.97 RIU^−1^, respectively. G. Mu et al. investigated the D-shaped MMF type of FSS and the influence of a polishing depth and length to sensor sensitivity; the maximum sensitivity of 10.57 RIU^−1^ can be achieved [29]. Although these works have made significant contributions to the measurement of RI using FSS, the influence of spatially modulated fiber speckle on RI measurement has not been explored yet. These investigations will promote a better comprehension of FSS performance, further optimize its effectiveness of accurately determining liquid analyte RI, and enable a profound insight into the interaction between fiber speckle patterns and the RI of liquid analytes. 

In this work, an FSS based on tapered multimode fiber (TMMF) to measure the RI of liquid analyte is fabricated. By the spatial modulation of the input beam into TMMF, the sensitivity of the FSS is enhanced significantly. The spatial modulation achieved by increasing the lateral and axial offset controls the excited high-order modes in the TMMF. Experiment results indicate that the more high-order modes are excited in the MMF, the higher is the sensor sensitivity exhibited by the FSS, which expresses a consistent tendency with the prediction. The variations of speckle induced by environment RI change around the taper region of the TMMF are quantified using a digital image correlations method called the zero-mean normalized cross-correlation coefficient. The sensor shows a high sensitivity of 20 RIU^−1^ and a resolution of 5.84 × 10^−5^ with good linear response to the variation of surrounding RI in the range of 1.3164 to 1.3588 at 1550 nm. Additionally, the speckle sensor possesses long-period temporal stability, with a digital image correlation variation of less than 0.003 within 600 s. Note that, compared to the work [26], we fixed the sensing head and changed the modes excitation, which maintains the stable structure of the sensing head. The various modes excitation are achieved by using the spatial modulation, which facilitates the control of the experimental system. The proposed RI FSS displays the advantages of a simple sensor module structure, efficient RI detection, and straightforward implementation.

## 2. Fiber Speckle Sensing Principle

### 2.1. Fiber Speckle Theory 

When a coherent laser emits light into an MMF, a speckle pattern will form at the output facet of the MMF. The transverse distribution of the speckle pattern is mainly affected by mode interference and mode coupling. The interferences of the modes are determined theoretically by the amplitudes and relative phases of each mode. Therefore, without considering the divergence and spatial diffraction of light from the output end of the fiber to the CMOS camera receiver, the light field *E*(*x*, *y*) and intensity distribution *I*(*x*, *y*) of the output speckle pattern projected on the CMOS receiving plane (*x*, *y*) can be expressed as:(1)$$E\left(x,y\right)=\sum_{k}A_{k}\exp\left(i\varphi_{k}\right)\Phi_{k}\left(x,y\right),$$
(2)$$I\left(x,y\right)=\sum_{m=1}^{k}\sum_{n=1}^{k}A_{m}A_{n}^{*}\exp\left[i\left(\varphi_{m}-\varphi_{n}\right)\right]\Phi_{m}\left(x,y\right)\Phi_{n}\left(x,y\right),$$
where *A_k_* and *φ_k_* represent the amplitude and relative phase. *m* and *n* are defined as the order of *m*-th and *n*-th eigenmodes Φ*_k_*. The intensity of the speckle pattern is highly influenced by the relative phase deviation *Δφ* = *φ_m_ − φ_n_*. The distribution of speckle intensity depends on the phase difference of modes and mode coupling effects. In this case, changing the RI of the liquid around the waist of the tapered fiber, we can achieve sensitive RI sensing by quantifying the deviation of the speckle pattern.

### 2.2. Mode Excitation in MMF

In the MMF with a step-index profile, the number of supported eigenmodes *M* are determined approximately by the normalized frequency *V*:(3)$$M\approx \frac{1}{2}V^{2}=\frac{1}{2}{\left(k_{0}a\sqrt{n_{1}^{2}-n_{2}^{2}}\right)}^{2},$$
where *k_0_*, *a*, *n*_1_, and *n*_2_ are the vacuum wavenumber, core diameter, core RI, and cladding RI, respectively. The excitations of modes are crucial for the complexity of speckle patterns. By increasing the number of excitation modes in MMF, the speckle pattern exhibits a more intricate intensity distribution. This enhanced complexity enables the sensor system to exhibit a heightened sensitivity in responding to external environmental RI variation. To increase the high-order modes in MMF, common methods focus on the circular symmetry disturbances of the waveguide structure. When the fundamental-mode Gaussian beam is launched into the input port of MMF, a lateral and axial offset between emitting SMF and MMF is introduced. it is an effective method.

In this work, to modulate the speckle pattern in space, the lateral and axial offset between SMF and MMF are adopted to high-order mode excitations. To compare the effects of different offset conditions on the mode excitation effect, power couplings of the first 88 modes excited in MMF to the SMF output light were computed theoretically. We considered both the mode overlap and the mismatch in the effective refractive index between modes. When computing, the light emitted from the SMF is considered as the fundamental mode Gaussian beam with an operation wavelength of 1550 nm, and the output facet of SMF is the beam waist. After propagating a short distance, which is less than the Rayleigh distance in free space, the Gaussian beam reaches the incident end face of the MMF, resulting in the excitation of the high-order modes. The core RI, cladding RI, core radius, and radius of MMF were set as 1.457, 1.442, 31.25 μm, and 62.5 μm, respectively.

In Figure 1, mode excitations of various spatial modulation conditions are exhibited, in which the *l* and *x* represent the axial and lateral offsets, respectively, between SMF and MMF; *N_total_* gives the number of excited modes. Spatial modulation can be achieved by changing the values of *l* and *x*. When the cores of SMF and MMF were aligned, i.e., *l* = 10 μm and *x* = 0 μm, with most of the power fraction focused on a small number of modes, 10 eigenmodes were excited. When *l* = 10 μm and *x* = 25 μm were introduced, the power distribution of the modes became uniform, and most of the high-order modes were allocated light power. Obviously, the number of excited high-order modes increases with the lateral offset. When the *l* remains constant and *x* was further increased to 30 μm, 59 eigenmodes were excited. However, the excitations of the high-order modes are limited by the large offset. When the *x* was increased continually to 35 μm, almost all modes, including low- and high-order modes, did not receive the power. Additionally, when the *l* = 55 μm, the number of excited modes increases slightly compared with the spatial modulation conditions of *l* = 10 μm and *x* = 30 μm. 

These results indicate that the adopting of offsets between SMF and MMF facilitates the excitation of high-order modes, and the lateral offset exerts a more considerable influence on the excitation of modes compared with the axial offset.

### 2.3. Digital Image Demodulation

The zero-mean normalized cross-correlation (ZNCC) value is applied to evaluate the correlation of the speckle pattern; it is a statistical image-matching method. The ZNCC can directly express the similarity between two images without requiring additional parameters. The original image captured by the CMOS camera contains the RGB information, and each speckle must be converted into a grayscale image before evaluating the correlation. The ZNCC shows the high robustness to linear and uniform brightness change by considering the subtraction of the grayscale mean values from both the reference and target images [30,31]. The light intensity represented by each pixel in the image contributes to the image correlation, and the general expression is as follows:(4)$$C_{ZNCC}=\frac{\sum_{N}\left[I_{0}\left(X,Y\right)-\bar{I_{0}}\right]\cdot \left[I_{K}\left(X,Y\right)-\bar{I_{K}}\right]}{\sqrt{\left[I_{0}\left(X,Y\right)-\bar{I_{0}}\right]\cdot \left[I_{K}\left(X,Y\right)-\bar{I_{K}}\right]}}$$
where *I*(*X*, *Y*) and *I* represent the light intensity at (*x*, *y*) point in the speckle image and the mean light-intensity value of the image. The subscripts 0 and *K* correspond to the reference speckle pattern and target speckle pattern, respectively. The output speckle pattern consists of *N* pixels.

## 3. Experimental Investigation and Results

### 3.1. Experimental Setup

Figure 2 exhibits the experimental setup of the FSS system. The coherent light with the operation wavelength of 1550 nm emitted by a fiber laser source (MC Fiber Optics, MCNLFL-1550-S-S2-0-FA-T1, DK Photonics, Shenzhen, China) was launched into an SMF (CORNING SMF-28e^®^) for alignment with MMF. It can avoid the impact of visible light in the environment on sensor performance. A laser with an output power of 10.09 mW was used for all studies, and there was no high saturation intensity in any experiment. An isolator (ISO) prevented back reflection and is used to protect source fiber lasers. 

A step index profile MMF (YOFC, SI 60/125-22/250, Yangtze Optical Fibre and Cable Joint Stock Limited Company (YOFC), Wuhan, China) was fabricated to a TMMF by a flame heating technique. Figure 3 shows the optical micrograph of TMMF. The diameter and length of the tapered area are 16 μm and 1.72 mm. The length of the full view field is 1.6 cm, as shown in Figure 3a. Figure 3b–d show the optical microscope images of the gradually tapered transition region, tapered fiber region, and gradually thickened transition region of TMMF, respectively.

The MMF was heated by an oxyhydrogen flame at 1300 °C and stretched by two rotary fiber holding blocks, with a constant speed and small reduction rate, for an adiabatic TMMF. Then, the free space couple between SMF and TMMF was achieved by the core alignment fusion splicer (Fujikura, Tokyo, Japan 88S+). The spatial modulation between the facets of SMF and MMF can be adjusted by the drive motor. The waist and transition areas of the tapered MMF were packaged in a 3D printed chamber. To reduce the environmental perturbations, the unstretched sections of TMMF were cured by the UV-curing adhesive (KSIMI, K2018), and two simplified fiber clamps (DHC Daheng Opitcs, GCX-M0104, Daheng New Epoch Technology, Inc., Beijing, China) were used to ensure the mechanical stability of the entire TMMF structure. The output port of TMMF was fixed on a manual XYZ translation stage, which is used to adjust the spatial location of output speckle patterns projected to the small-sized infrared CMOS camera (CINOGY, CinCam CMOS Series, CMOS-1201-IR, CINOGY TECHNOLOGIES, Duderstadt, Germany, 40 × 40 × 20 mm^3^). The camera has a damage threshold of 100 mW/cm^2^, a resolution of 1280 × 1024, and a dynamics range of 43 dB, which can satisfy the brightness and contrast requirements. 

Temperature variations could change speckle patterns, and relevant works have reported the use of FSS for temperature responses [22,24]. To avoid the impact of temperature on the RI measurement results, we maintain the room temperature at 25 °C. All the experiments operated at room temperature, and the sensor system was fixed on the optical breadboard to reduce the impact of the environment.

### 3.2. Mode Excitation and RI Measurement

To achieve variation of RI measurement by using ZNCC, mode excitation and modulation of modes by evanescent field are indispensable. Tapered fibers can generate strong evanescent fields, and the power of guided modes in tapered fibers propagates in evanescent waves and accounts for a substantial proportion; the effective refractive index of the guided mode is easily modulated by the external medium RI. We employed an effective and common approach to increase the high-order modes by varying the lateral offset *x* and axial offset *l* when spatially modulating between the SMF and MMF; the output speckle patterns with varying offset conditions are given. Figure 4 depicts the optical micrograph of *l* = 10, 25, and 60 μm without lateral offset, as well as its corresponding speckle patterns. The ZNCC of Figure 4d–f are 0.89, 0.82, and 0.96, respectively.

Figure 5 exhibits the optical micrograph with different lateral offsets and their corresponding speckle patterns when maintaining the axial offset and shows that the lateral and axial offsets are increased simultaneously. Figure 5a shows the direct core-to-core coupling between SMF and MMF, with only *l* = 10 μm and *x =* 0. In addition, the corresponding speckle pattern, given in Figure 5e, was captured when the environmental liquid around the waist region of tapered MMF is water, whose RI is 1.3164 at 1550 nm. Figure 5b exhibits the spatial modulation conditions of *l* = 10 μm and *x* = 17 μm. Figure 5c further increases the *x* to 26 μm. The ZNCC of Figure 5e–g is 0.50, 0.51, and 0.54, respectively. The smaller the value shown, the bigger the spatial distribution variation became. Compared with Figure 4d–f and Figure 5e,f, the ZNCC induced by the lateral offsets are greater than those induced by the axial offset, which indicates the lateral offset can modulate the spatial distribution of the speckle pattern efficiently. If the lateral offset were further expanded slightly, the output speckle pattern almost disappears. So, the lateral offset threshold is considered as 30 μm. The axial offset *l* is increased to 55 μm, the optical micrograph is shown in Figure 5d, and the corresponding speckle pattern is exhibited in Figure 5h.

These results indicate that the presence of offsets between SMF and MMF can promote the excitation of high-order modes, and the lateral offset plays a dominant role in the mode excitation compared to the axial offset, which is consistent with the theoretical calculations.

Glycerol–water solutions with different concentrations were applied to calibrate and measure the RI value because glycerol is nontoxic and less volatile, and it can be mutually dissolved with water in any concentration ratio. By adjusting the volume percentage of glycerol, the RI range of the solution was calibrated from 1.3164 to 1.3588 [32]. The solutions with different RI were added to the chamber that packages the TMMF; the output speckle pattern was captured and the image correlation between them was evaluated using ZNCC. In Figure 6, speckle patterns with various external RI and a high-quality linear response of ZNCC with an RI variation from 1.3164 to 1.3588 are displayed experimentally. The sensitivity of FSS, the absolute value of the linear curve slope, is 6.41 RIU^−1^.

### 3.3. Investigation of FSS Sensitivity 

In theory, speckle patterns are developed by the coherent superposition of guided modes. By increasing the number of guided modes, the complexity of speckle patterns can be enhanced, leading to an improvement in the sensitivity of FSS.

In these different SMF and MMF spatially modulated states mentioned in Figure 4a–c, we measured the responses to RI variation with corresponding spatial modulation conditions, and their FSS performance and offset conditions are shown in Figure 7a. When the axial offset is 10, 25, and 60 μm, the corresponding sensitivities of FSS are 7.6, 9.1, and 9.6 RIU^−1^, respectively. Figure 7b exhibits the performance of FSS under the varying spatial modulation conditions described in Figure 5a–d. The comparable sensitivity of FSS is 6.5, 7.8, and 11.4 RIU^−1^ for lateral offsets of 17, 26, and 30 μm, respectively. Most significantly, FSS had a sensitivity of 19.52 RIU^−1^ when the axial and lateral offsets were set to 30 and 55 μm, respectively. The increase in spatial offset between SMF and MMF promotes the improvement of FSS sensitivity. 

## 4. Discussion

To ascertain the reliability of RI FSS, we carried out numerical simulation of the responses based on various modes to RI change by using the software OptiBPM 13.1.3. The results are exhibited in Figure 8. The inset of Figure 8 depicts the simulated structure and the related area’s RI distribution. In the layout, the RI of yellow core and green cladding were set as 1.457 and 1.442, respectively. The RI of blue liquid analyte was changed from 1.3301 to 1.3601. The final simulation results were obtained using the 5 and 20 LP modes as the red input fields depicted in Figure 8. The change in speckle pattern correlation brought on by the same change in RI will steadily increase as the number of conduction modes rises, which is consistent with the experimental investigation of Figure 7b.

Through the optimization of the sensor structure and the enhancement of evanescent TMMF, it may be possible to achieve the FSS potential for improved sensitivity.

To study the variation of ZNCC value and FSS system error, temporal stability of the sensor system in air and water were measured in 600 s, respectively. The output speckle patterns were recorded at every time interval of one second, approximately. An evolution sequence of speckle pattern variation over time was obtained by making a ZNCC correlation computing for each recorded image with the initial speckle pattern.

As shown in Figure 9, the speckle correlation obtained by placing the sensor head directly in air changes insignificantly with time. The ZNCC value reduced from 1 to 0.997 within 600 s. All data are within the confidence intervals of plus or minus 3.5-times standard deviation (σ = 0.00351). The system error is caused by the combination between the intrinsic electronic noise of the camera, environment noise, and stability shift of the laser source (power variation ≤ 0.05 dB in 15 min), which influence the discrepancies of ZNCC for sensing. The maximum decline value of the speckle correlation variation, observed when the TMMF was immersed in water, is slightly larger than that obtained in air. Only the two sampling points are outside the confidence intervals of plus or minus 3.5-times the standard deviation. The measurement time for a single experiment is less than 10 min, and the overall data are relatively stable. We conclude that for RI detection, the FSS system is effective and robust. 

Previous works have demonstrated various approaches for evaluating the resolution of FSS. One of them presented a ratio between defined minimum ZNCC variation (0.01) and sensitivity [27]. One method thought the conservative resolution interval was 10 times the standard deviation of sampled data during the 10 s [28], and the other one computed a ratio of 3 times the standard deviation and sensitivity in a blank sample measurement [29]. The ZNCC value is an evaluation metric for the correlation of two images, and the grayscale value of each pixel in the image contributes to the total ZNCC value. Theoretically, if there is a grayscale variation in one pixel, it will cause a change in ZNCC value. However, there are almost no scenes where one grayscale value in one pixel changes in actual experiments. The change in the ZNCC value caused by the sensor system’s error will be greater than in this case. The ZNCC change corresponding to the resolution needs to be larger than that of the system error. The mentioned evaluating method of resolution in [28] is considered because it minimizes the influence of time as much as possible. To achieve the optimal resolution, the multiple measurements on the state when the sensing head was immersed in water were operated. As shown in Figure 10, the image correlation between each measured speckle pattern and the previous measured speckle pattern is given. 

The maximum variation of ZNCC caused by the number of measurements is 0.0012, and the relative standard deviation (RSD) of the data is 0.00038. So, the maximum critical value of resolution of the current FSS can be defined as 5.84 × 10^−5^ RIU in a group of experiments. Only when the change of external RI is greater than this value can the ZNCC detect the variation. When the FSS sensitivity of 19.52 RIU^−1^ is considered, the minimum resolution was evaluated as 5.84 × 10^−5^ RIU. 

To compare the performances of previously reported RI FSS, Table 1 summarizes the typical parameters of RI FSS in recent years, including sensor head, sensitivity, resolution, and response range. Compared with other approaches, the TMMF method is simple and suitable for sensitive detection. 

Note that the ZNCC can determine the difference in correlation between two images but limits determining the specific values of the parameters that cause image correlation changes. However, the ZNCC shows the equivalence relation with other relevant criteria such as a zero-mean normalized sum of squared difference (ZNSSD) and parametric sum of squared difference (PSSD) criterion with two additional unknown parameters (PSSD_ab_), which can simplify theoretical analysis and practical application [33]. Further investigation can concentrate on real-time demodulation of RI, which could be achieved through built-in integrated algorithms.

## 5. Conclusions

In conclusion, this paper presents an experimental investigation of a fiber speckle sensor based on a TMMF. The sensor was assessed by measuring different ratios of glycerol–water solution and obtained a sensitivity of 19.52 RIU^−1^ and a resolution of 5.84 × 10^−5^ over a linear response range of 1.3164 to 1.3588 refractive index. The experiment demonstrated that the sensitivity of the speckle sensor is affected by the number of propagation modes. The results indicate that the fiber speckle sensor provides a simple, efficient, and economic approach to fiber sensing, which allows for a reliable alternative for RI detection. This work has important implications for expanding fiber optic sensing technology and has potential applications in fields such as oceanography research, ocean environment monitoring, and ocean fisheries. Further investigations into the FSS’s high sensitivity and wide linear response range may focus on non-adiabatic TMMF and the combination with other techniques.

## Figures and Tables

**Figure 1 sensors-23-06814-f001:**
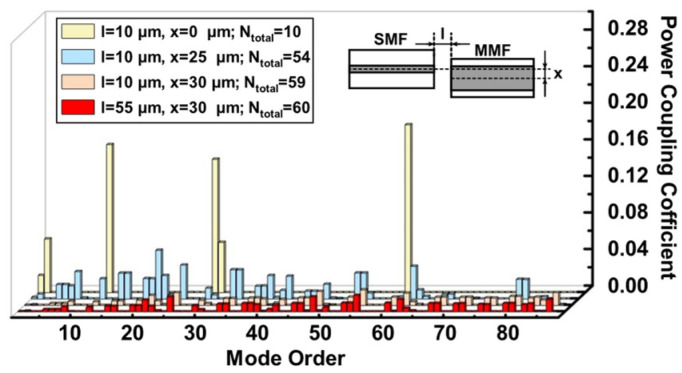
The mode power coupling between SMF and MMF with different axial and lateral offsets.

**Figure 2 sensors-23-06814-f002:**
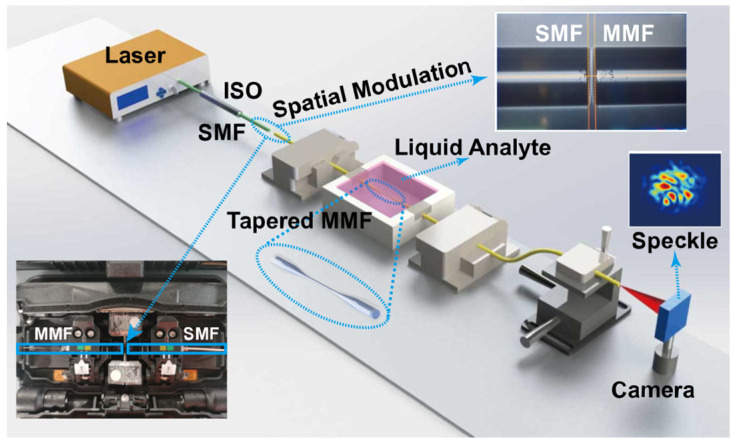
Configuration of the experimental RI measurement setup. Single-mode fiber (SMF); multimode fiber (MMF).

**Figure 3 sensors-23-06814-f003:**
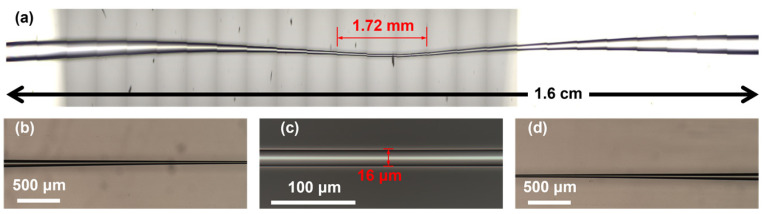
Optical micrograph of a TMMF: (**a**) the full view field of TMMF; (**b**) gradually tapered transition region; (**c**) tapered region; (**d**) gradually thickened transition region.

**Figure 4 sensors-23-06814-f004:**
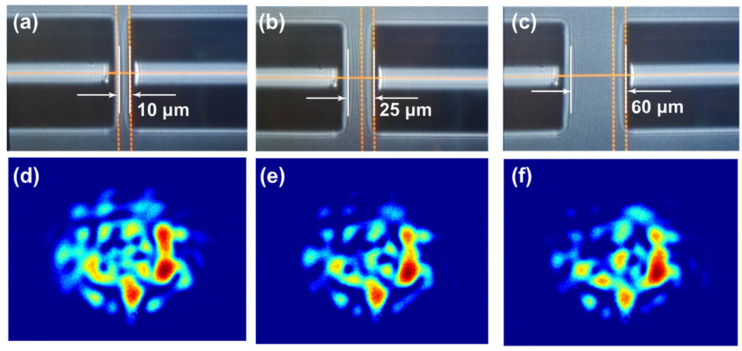
The optical micrograph when SMF and MMF only have an axial offset and its corresponding speckle pattern, the environmental RI is 1.3164. (**a**) *l* = 10 μm; (**b**) *l* = 25 μm; (**c**) *l* = 60 μm; (**d**–**f**) the output speckle patterns corresponding.

**Figure 5 sensors-23-06814-f005:**
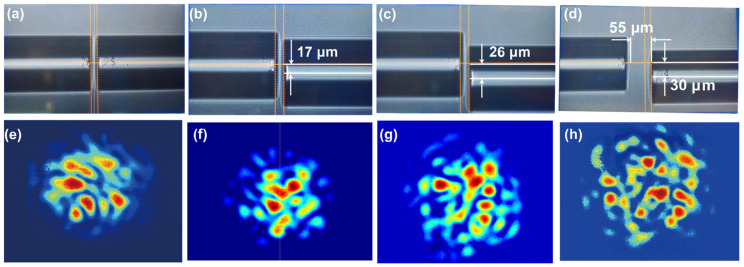
The optical micrograph of spatial coupling between SMF and MMF under different offsets: (**a**) *l* = 10 μm and *x =* 0; (**b**) *l* = 10 μm and *x* = 17 μm; (**c**) *l* = 10 μm and *x* = 26 μm; (**d**) *l* = 55 μm and *x* = 30 μm; (**e**–**h**) the output speckle patterns obtained when external RI is 1.3164 under the corresponding (**a**–**d**) offset states.

**Figure 6 sensors-23-06814-f006:**
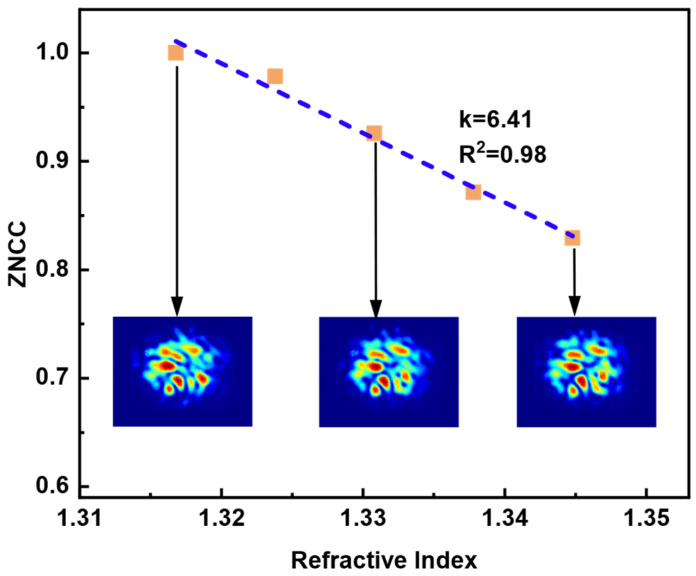
The value of ZNCC with the variation of the different RI. A linear response curve displays a 6.41 RIU^−1^ sensitivity.

**Figure 7 sensors-23-06814-f007:**
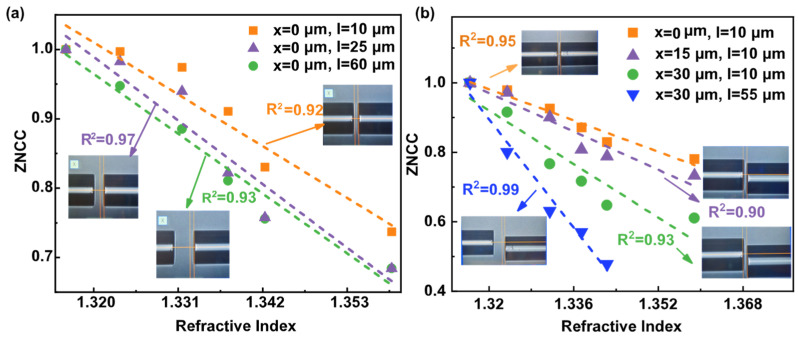
The linear response of ZNCC with RI under three offset conditions when SMF and MMF are spatially coupled; (**a**) FSS response with increased axial offset; (**b**) FSS response with increased lateral offset, as well as increased lateral and axial offset.

**Figure 8 sensors-23-06814-f008:**
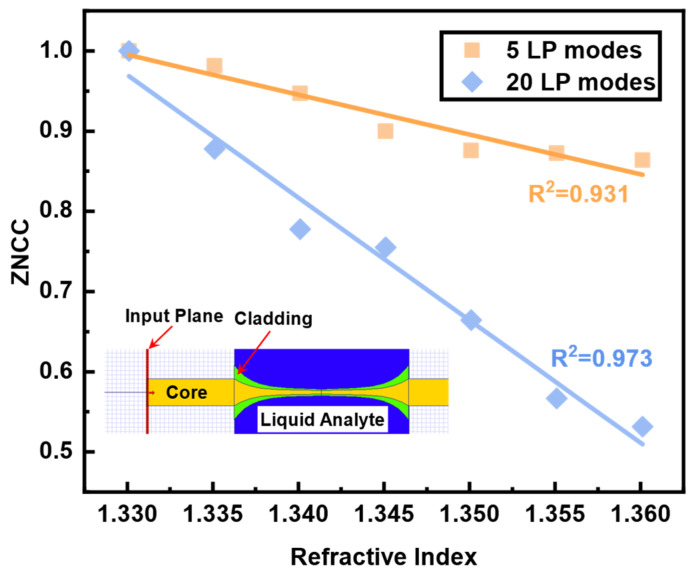
The response of the number of LP modes to RI FSS; inset is the simulation layout.

**Figure 9 sensors-23-06814-f009:**
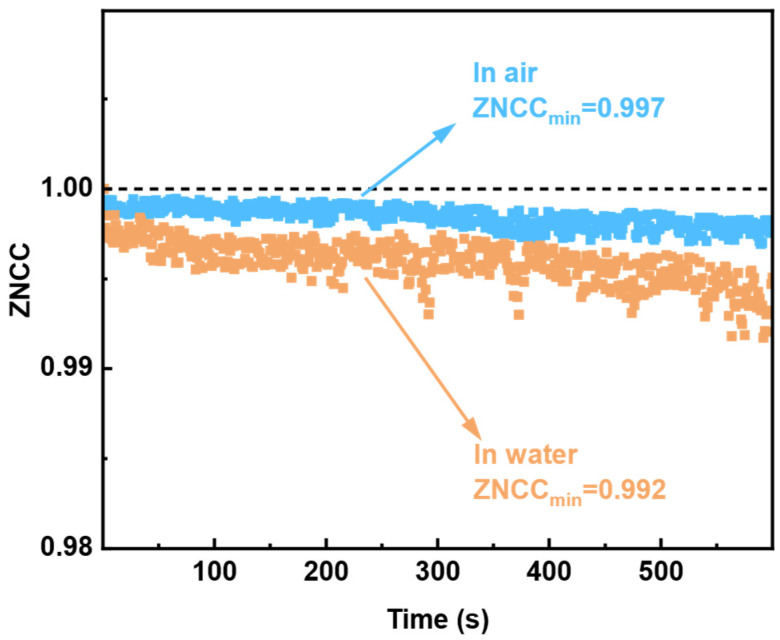
The experiment results of the FSS system’s temporal stability; the taper region of TMMF is immersed in air and water.

**Figure 10 sensors-23-06814-f010:**
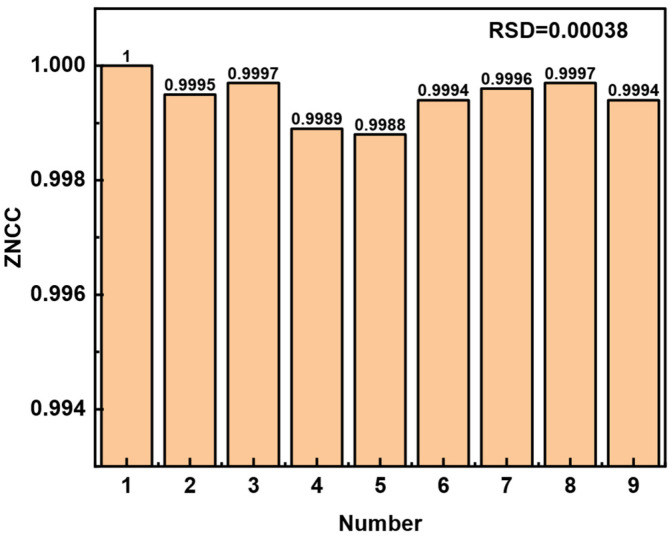
Multiple measurements of the state when sensing head is immersed in water.

**Table 1 sensors-23-06814-t001:** Sensing performance comparison of FSS with different sensing heads for RI measurement.

Sensor Head	Sensitivity (RIU^−1^)	Resolution (RIU)	Response Range	Ref.
MMF-NCF-MMF	18.70	5.35 × 10^−4^	1.33–1.36	[27]
Exposed MMF	10.97	4.60 × 10^−4^	1.33–1.375 Nonlinear	[28]
D-Shaped MMF	10.57	2.12 × 10^−5^	1.40–1.44	[29]
TMMF	19.52	5.84 × 10^−5^	1.3164–1.3588	This work

## Data Availability

Not applicable.

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
