# Peer review of "Spatially Modulated Fiber Speckle for High-Sensitivity Refractive Index Sensing"

_sensors, 2023, doi:10.3390/s23156814_

Round 1
Reviewer 1 Report
The manuscript presents a tapered MMF as RI sensing structure with different higher order modes excited via axial and lateral offsets between SMF and MMF. The interrogation is based on so called “zero mean normalized cross-correlation coefficient (ZNCC)” which is the measure of pixelwise relative intensity variation in the output fiber speckle patterns which change as the evanescent field interacts with the RI medium in the TMMF section. In my opinion, the paper seems like an attempt to publish a “quick paper” without offering any innovation or substantial merits or broad theoretical and experimental analysis. The use of TMMF structure with axial and lateral core offsets for different no. of higher order modes excitations is very basic concept and the method of ZNCC is nothing new either. The study lacks theoretical/simulation backing with very limited data on experiments and insufficient background information while discussing the limitations of the interrogation method. Without further theoretical/simulation data support and broader experimental data and analysis, I would not recommend publishing this paper in sensor Journal.
However, if the authors address the following concerns, I expect the quality of the work would improve, for it to be meaningful to the community in the fiber sensor technology.
1. For a paper of 18 co-authors, I would expect someone do a theoretical simulation study and calculate the ZNCC values for speckle patterns as they change for different axial and lateral offsets and no. of modes excited. The theoretical study adds value to the paper when analyzing the holistic picture including limitations of the interrogation method. Then I would recommend comparing the experimental results with the simulation data.
2. When you compare the relative intensity of pixels, one can expect the saturation of those pixels beyond certain intensity and the laser power. Beyond which grayscale image would not be that effective to resolve the intensity difference unless you have high resolution camera with high dynamic range of pixels.
Authors neither mention the power of the laser used nor the specifications of the CMOS camera pixels. Please provide this specification and discuss in detail the consequences/limitations of the interrogation method in appropriate section in the manuscript.
3. The Fujikura fusion splicer looks like part of the set-up itself, not used for fusion splicing but to control and maintain offsets between SMF and MMF. If so, it has to be the part of set-up and has to be depicted as such in the schematic of Figure 2. The optical isolator has been used but not depicted in the schematic either. Please revise the schematic with the complete components needed.
The free-space part of the set-up for lateral offset is not that attractive part of the set-up. I would suggest authors perform experiments in a systematic way to differentiate the isolated effect of axial and lateral offsets.
i.e.
a) Keep lateral offset zero (x= 0) and vary the axial offset (parameter ‘L’).
b) Keep axial offset the same (L constant) and vary the lateral offset x. You may do several of them for statistical correlation analysis.
c) Include the case for SMF and MMF fusion spliced (L=0) and different lateral offsets values. One can control excitation of higher order modes with lateral offsets alone too. Getting rid of the free space coupling part is important for any practical application of the sensor for its reliability.
4. Please specify the dimensions of the tapered MMF and include an optical micrograph of the same.
5. The fiber speckle theory needs a deeper dive that can include the simulation produced speckle patterns and impact of the offsets and other parameters such as fiber core diameters and the no. of excited modes etc.
6. The ZNCC method needs a bit more succinct elaboration and its limitations or advantages.
7. I am not sure whether the reported RI values here are for 1550 nm wavelength. If yes, please briefly mention the calibration method used and include the table with % glycerol solutions and their RI values measured. If the measured RI values are for 589.3 nm which is the wavelength several refractometer use, see the paper below to estimate the RI values at 1550 nm. I would suggest authors include higher RI values, even though the response might be no-linear as the RI of the liquid approaches the fiber core-index. one can easily prepare glycerol-based solution with higher RI values up to 1.4571.
D. Karki et al "Optimization of the fourth self-imaging spectral response in magnetic fluid cladded-MMI fiber optic sensor for magnetometry," Proc. SPIE, (2023)
https://doi.org/10.1117/12.2664107.
8. In line 258-259, authors mention “All data is within the confidence intervals of plus or minus 3.5 times standard deviation”.
This statement has no specific meaning in statistics other than referring that the data has normal distribution. In any normal distribution of data, 99.7 % of data lie within 3 standard deviations. Please specify the standard deviation first to see how close the measured ZNCC values are over the time to see its reliability. More correlation and regression statistical analysis can be added too.
9. Please show how you calculated the resolution of RI that can be detected with the sensor. It must be related to the ZNCC resolution which in turn should be directly related to specification of the camera pixels not mentioned anywhere in the text. Please provide all relevant information and specifications on which sensor performance metrics depend on.
10. In line 263-264, the extra evaporation of the liquid has been blamed for some discrepancies. First the liquid is at room temperature. Second, why don’t authors use deep enough column of the water/liquid so that the sensor is not impacted by the evaporation at surface of the liquid. A few millimeters deep liquid column immersion of the sensor is enough to avoid any doubt due to evaporation.
11. There are several sentences that need restructuring/ rephrasing such as sentences in line 65-66, line 194-196, line 278-279 … etc.
There are several sentences that need restructuring/ rephrasing such as sentences in line 65-66, line 194-196, line 278-279 … etc. Needs careful proof reading and corrections.
Author Response
Reviewer 1
The manuscript presents a tapered MMF as RI sensing structure with different higher order modes excited via axial and lateral offsets between SMF and MMF. The interrogation is based on so called “zero mean normalized cross-correlation coefficient (ZNCC)” which is the measure of pixelwise relative intensity variation in the output fiber speckle patterns which change as the evanescent field interacts with the RI medium in the TMMF section. In my opinion, the paper seems like an attempt to publish a “quick paper” without offering any innovation or substantial merits or broad theoretical and experimental analysis. The use of TMMF structure with axial and lateral core offsets for different no. of higher order modes excitations is very basic concept and the method of ZNCC is nothing new either. The study lacks theoretical/simulation backing with very limited data on experiments and insufficient background information while discussing the limitations of the interrogation method. Without further theoretical/simulation data support and broader experimental data and analysis, I would not recommend publishing this paper in sensor Journal.
However, if the authors address the following concerns, I expect the quality of the work would improve, for it to be meaningful to the community in the fiber sensor technology.
- For a paper of 18 co-authors, I would expect someone do a theoretical simulation study and calculate the ZNCC values for speckle patterns as they change for different axial and lateral offsets and no. of modes excited. The theoretical study adds value to the paper when analyzing the holistic picture including limitations of the interrogation method. Then I would recommend comparing the experimental results with the simulation data.
A: Thanks for your advice. As suggested by the reviewer, we performed additional simulation and analysis of the simulation results. As shown in Figure 8, the simulation findings of the responses of various modes to RI change are exhibited. The comparison of simulation results and experimental measurements makes a strong case that the change in speckle pattern correlation brought on by the same change in RI will steadily increase as the number of conduction modes rises.
The simulated modeling is used here to qualitatively account for the impact of mode number on FSS sensitivity rather than to provide an accurate quantitative comparison with the experiment. The completed factors such as the temperature and noise will be considered in future. [The detail descriptions are given in revised manuscript line 276-287.]
Figure 8. the response of the number of LP modes to RI FSS, inset is the simulation layout.
- When you compare the relative intensity of pixels, one can expect the saturation of those pixels beyond certain intensity and the laser power. Beyond which grayscale image would not be that effective to resolve the intensity difference unless you have high resolution camera with high dynamic range of pixels. Authors neither mention the power of the laser used nor the specifications of the CMOS camera pixels. Please provide this specification and discuss in detail the consequences/limitations of the interrogation method in appropriate section in the manuscript.
A: Thanks for reviewer’s comment. For the saturation limits of maximum CW power, the large beam diameter shows the endurance for the large laser power. In our experiment, a laser with an output power of 10.09 mW was used for all studies, and there was no high saturation intensity in all experiment. The camera has a damage threshold of 100 mW/cm2, a resolution of 1280 × 1024, and a dynamic range of 43 dB, which can satisfy the brightness and contrast requirements. [The description is added to in revised manuscript line 201-203.]
- The Fujikura fusion splicer looks like part of the set-up itself, not used for fusion splicing but to control and maintain offsets between SMF and MMF. If so, it has to be the part of set-up and has to be depicted as such in the schematic of Figure 2. The optical isolator has been used but not depicted in the schematic either. Please revise the schematic with the complete components needed.
A: Thanks for reviewer’s advice. As suggested by reviewer, the fusion splicer and ISO have been added in the Figure 2.
Figure 2. Configuration of the experimental RI measurement setup. single mode fiber (SMF); multimode fiber (MMF); isolator (ISO).
- The free-space part of the set-up for lateral offset is not that attractive part of the set-up. I would suggest authors perform experiments in a systematic way to differentiate the isolated effect of axial and lateral offsets.
i.e.
- a) Keep lateral offset zero (x= 0) and vary the axial offset (parameter ‘L’).
- b) Keep axial offset the same (L constant) and vary the lateral offset x. You may do several of them for statistical correlation analysis.
- c) Include the case for SMF and MMF fusion spliced (L=0) and different lateral offsets values. One can control excitation of higher order modes with lateral offsets alone too. Getting rid of the free space coupling part is important for any practical application of the sensor for its reliability.
A: We appreciate the reviewer’s comment We have provided experiments on the spatial modulation state and its impact on FSS sensitivity. The optical micrograph and corresponding speckle pattern of the recommended schemes are shown in Figures 4 and 5 [The description is added to in revised manuscript line 213-227 and 233-237.]. The FSS performances with various offset conditions are also supplied in Figure 7. [The description is added to in revised manuscript line 268-273.]
Figure 4. The optical micrograph when SMF and MMF only have axial offset, and its corresponding speckle pattern, the environmental RI is 1.3164. (a) l=10 μm; (b) l=25 μm; (c) l=60 μm; (d)-(f) the output speckle patterns corresponding (a)-(c) offset states.
Figure 5. The optical micrograph images of spatial coupling between SMF and MMF under different offsets: (a) l=10 μm and x=0; (b) l=10 μm and x= 17 μm; (c) l=10 μm and x=26 μm; (d) l=55 μm and x= 30 μm; (e)-(h) the output speckle patterns obtained when external RI is 1.3164 under the corresponding (a)-(d) offset states.
Figure 7. The linear response of ZNCC with RI under three offset conditions when SMF and MMF are spatially coupled. (a) FSS response with increased axial offset; (b) FSS response with increased lateral offset, as well as increased lateral and axial offset.
- Please specify the dimensions of the tapered MMF and include an optical micrograph of the same.
A: The optical micrograph schematic diagrams of TMMF are given in Figure R6 (Figure 3 in the manuscript). Figure 3 shows the optical micrograph of TMMF. The diameter and length of the tapered area are 16 μm and 1.72 mm. The length of full view field is 1.6 cm shown in Figure 3a. Figure 3b-d give the optical microscope of gradually tapered transition region, tapered fiber region and gradually thickened transition region of TMMF, respectively. [The description is added to in revised manuscript line 183-189.]
Figure R6. Schematic diagram of a TMMF pulled down by an optical microscope, (a) the full view field of TMMF; (b) gradually tapered transition region; (c) tapered region; (d) gradually thickened transition region.
- The fiber speckle theory needs a deeper dive that can include the simulation produced speckle patterns and impact of the offsets and other parameters such as fiber core diameters and the no. of excited modes etc.
A: Thanks for the reviewer’s comment. We added the numerical simulations of impact on mode number M and the FSS sensitivity. Studying the impact of waveguide structure such as core diameter on FSS sensitivity can be reflected by the influence of modes on sensitivity. Based on the Equation 3 in revised manuscript, the core diameter of the step-index profile MMF directly affects the modes in the fiber, and the superposition of modes is the direct cause of speckle pattern.
(3)
- The ZNCC method needs a bit more succinct elaboration and its limitations or advantages.
A: The ZNCC can determine the difference in correlation between two images but limits in determining the specific values of the parameters that cause image correlation changes. However, besides the high robustness to linear and uniform brightness change mentioned in the manuscript, the ZNCC can directly express the similarity between two images without requiring additional parameters. In addition, ZNCC shows the equivalence relation with other relevant criteria such as a zero-mean normalized sum of squared difference (ZNSSD) and parametric sum of squared difference (PSSD) criterion with two additional unknown parameters (PSSDab), which can simplify theoretical analysis and practical application [R1]. [The description is added to in revised manuscript line 156-158 and 340-345.]
- I am not sure whether the reported RI values here are for 1550 nm wavelength. If yes, please briefly mention the calibration method used and include the table with % glycerol solutions and their RI values measured. If the measured RI values are for 589.3 nm which is the wavelength several refractometer use, see the paper below to estimate the RI values at 1550 nm. I would suggest authors include higher RI values, even though the response might be no-linear as the RI of the liquid approaches the fiber core-index. one can easily prepare glycerol-based solution with higher RI values up to 1.4571.
- Karki et al "Optimization of the fourth self-imaging spectral response in magnetic fluid cladded-MMI fiber optic sensor for magnetometry," Proc. SPIE, (2023) https://doi.org/10.1117/12.2664107.
A: Thanks for reviewer’s suggestions, the RI values at 1550 nm have been calibrated by using provided reference and horizontal axis of Figure 6 and Figure 7 in the manuscript have been revised. And the mentioned paper is citated as Ref. 33 in the manuscript. In our previous work, we have tried the higher RI sensing such as beyond 1.36 [R3]. The experiment results characterize great linearity, which indicates reliable accuracy.
- In line 258-259, authors mention “All data is within the confidence intervals of plus or minus 3.5 times standard deviation”. This statement has no specific meaning in statistics other than referring that the data has normal distribution. In any normal distribution of data, 99.7 % of data lie within 3 standard deviations. Please specify the standard deviation first to see how close the measured ZNCC values are over the time to see its reliability. More correlation and regression statistical analysis can be added too.
A: As a common method in data processing, the μ±3.5σ shows the high accuracy on evaluating reliability and adopt in sensors area [R2]. The extra correlation and regression statistical analysis are not calculated exactly. However, we calculate the standard deviation of 0.0035 and included it in the manuscript to satisfy the reviewer’ request.
- Please show how you calculated the resolution of RI that can be detected with the sensor. It must be related to the ZNCC resolution which in turn should be directly related to specification of the camera pixels not mentioned anywhere in the text. Please provide all relevant information and specifications on which sensor performance metrics depend on.
A: The change of RI induced by the environmental noise can be several 10-5 RIU, which is beyond the RI variation corresponding to the change per grayscale per pixel. The ideal RI variation is practically submerged by the experimental situation. So, we adopt the mentioned parameter in the Table 1 in manuscript for the evaluation of RI resolution.
- In line 263-264, the extra evaporation of the liquid has been blamed for some discrepancies. First the liquid is at room temperature. Second, why don’t authors use deep enough column of the water/liquid so that the sensor is not impacted by the evaporation at surface of the liquid. A few millimeters deep liquid column immersion of the sensor is enough to avoid any doubt due to evaporation.
A: The reviewer’s viewpoint is right. The liquid column immersion of sensor in our experiment is in few millimeters (3 mm). And the impact on evaporation can be ignored as the reviewer’s highlight. We remove this description in the manuscript line 306-307. The system error is caused by the combination between the intrinsic electronic noise of the camera, environment noise and stability shift of the laser source (power variation ≤ 0.05 dB in 15 minutes), which influence the discrepancies of ZNCC for sensing.
- There are several sentences that need restructuring/ rephrasing such as sentences in line 65-66, line 194-196, line 278-279 … etc.
A: We have rephrased the sentences as reviewer’s request and checked the grammar mistakes in the whole manuscript. [The description is added to in revised manuscript line 69-71, 207-209 and 302-321.]
[R1] Pan, B. (2011). Recent progress in digital image correlation. Experimental mechanics, 51, 1223-1235.
[R2] Cabral, T. D., Fujiwara, E., Warren-Smith, S. C., Ebendorff-Heidepriem, H., & Cordeiro, C. M. (2020). Multimode exposed core fiber specklegram sensor. Optics letters, 45(12), 3212-3215.
[R3] P. Guo et al., (2022) Refractive Index Detection of Liquid Analyte in Broad Range Using Multimode Fiber Speckle Sensor, 2022 Asia Communications and Photonics Conference (ACP), Shenzhen, China, 2022, pp. 1981-1983, doi: 10.1109/ACP55869.2022.10088535.

Reviewer 2 Report
Review of Sensors [sensors-2488792]
Title: Spatially modulated fiber speckle for high-sensitive refractive index sensing
Authors: P. Guo et al.
This paper has presented an interesting refractive index (RI) sensor based on fiber speckle pattern imaging. The work showed that the speckle pattern images recorded with a CMOS camera placed at the end of the tapered optical fiber (TOF) are highly sensitive to the RI of the medium to be measured. Furthermore, the effect of the axial and lateral offset of input light into TOF on the sensor sensitivity was experimentally investigated. The paper can be accepted provided that comments raised could be addressed.
1. This work seems quite similar to the latest paper (F. Ari et al, “Tapered fiber optic refractive index sensor using speckle pattern imaging,” Optical Fiber Technology 79 (2023)) in terms of sensor layout, its working principle, RI sensitivity, and RI measurement range. Please explain about the technical improvement in your sensor compared to the previous work in the revised manuscript.
2. Is a special reason for using the 1550 nm NIR laser in this experiment? Because it needs the expensive infrared camera and decrease V number related to number of modes. I think that the visible laser such as He-Ne laser would be a good choice for low-cost and increase in V number, resulting in the increase in speckle complexity.
3. If possible, please a microscope image of TMMF showing the tapered region including TMMF transition, waist areas. Moreover, mention about TMMF waist length, waist diameter, and transition length.
4. Please discuss about issue and limitation of current sensor platform to be addressed.
The paper is well written in English.
Author Response
Reviewer 2
Review of Sensors [sensors-2488792]
Title: Spatially modulated fiber speckle for high-sensitive refractive index sensing
Authors: P. Guo et al.
This paper has presented an interesting refractive index (RI) sensor based on fiber speckle pattern imaging. The work showed that the speckle pattern images recorded with a CMOS camera placed at the end of the tapered optical fiber (TOF) are highly sensitive to the RI of the medium to be measured. Furthermore, the effect of the axial and lateral offset of input light into TOF on the sensor sensitivity was experimentally investigated. The paper can be accepted provided that comments raised could be addressed.
- This work seems quite similar to the latest paper (F. Ari et al, “Tapered fiber optic refractive index sensor using speckle pattern imaging,” Optical Fiber Technology 79(2023)) in terms of sensor layout, its working principle, RI sensitivity, and RI measurement range. Please explain about the technical improvement in your sensor compared to the previous work in the revised manuscript.
A: We much appreciate the reviewer for reminding us of the recent publication on Optical Fiber Technology 79 (2023). We have added this reference in the revised manuscript [R3]. Compared to this publication, we fixed the sensing head and changed the modes excitation, which maintains the stable structure of sensing head. The various modes excitation achieved by using the spatial modulation, which facilitates control of the experimental system. And the simulated modeling is used to qualitatively account for the impact of mode number on FSS sensitivity. [The description is added to in revised manuscript line 91-94]
The publication on Optical Fiber Technology 79 (2023) studied the core diameter influence of FSS performance. The waveguide structure of sensing head is changed, which causes the complexity of taper fiber fabrications and limits the consistence for each experiment.
- Is a special reason for using the 1550 nm NIR laser in this experiment? Because it needs the expensive infrared camera and decrease V number related to number of modes. I think that the visible laser such as He-Ne laser would be a good choice for low-cost and increase in V number, resulting in the increase in speckle complexity.
A: Thanks for reviewer’s viewpoint, The He-Ne laser would increase in V number for used MMF. However, 1550 nm is in the communication windows band, using 1550 nm laser may show rich application scenarios and potential. And it can avoid the impact of visible light in the environment on sensor performance. [The description is added to in revised manuscript line 177]
- If possible, please a microscope image of TMMF showing the tapered region including TMMF transition, waist areas. Moreover, mention about TMMF waist length, waist diameter, and transition length.
A: As suggestion by reviewer, the optical micrograph schematic diagrams of TMMF are given in Figure R6 (Figure 3 in the manuscript). Figure 3 shows the optical micrograph of TMMF. The diameter and length of the tapered area are 16 μm and 1.72 mm. The length of full view field is 1.6 cm shown in Figure 3a. Figure 3b-d give the optical microscope of gradually tapered transition region, tapered fiber region and gradually thickened transition region of TMMF, respectively. [The description is added to in revised manuscript line 183-189.]
Figure 3. Optical micrograph of a TMMF, (a) the full view field of TMMF; (b) gradually tapered transition region; (c) tapered region; (d) gradually thickened transition region.
- Please discuss about issue and limitation of current sensor platform to be addressed.
A: ZNCC is an image matching method that can only directly measure changes in the correlation of speckle pattern, while changes in RI need to be achieved through image correlation demodulation. Further investigation can concentrate on real-time demodulation of RI, which could be achieved through built-in integrated algorithms. [The relevant descriptions have been added to the manuscript line 346-348].
[R3] P. Guo et al., (2022) Refractive Index Detection of Liquid Analyte in Broad Range Using Multimode Fiber Speckle Sensor, 2022 Asia Communications and Photonics Conference (ACP), Shenzhen, China, 2022, pp. 1981-1983, doi: 10.1109/ACP55869.2022.10088535.

Author Response
Reviewer 3
Review for the article Spatially Modulated Fiber Speckle for High-sensitivity Refractive Index Sensing, presented for publishing in the journal Sensors. This paper proposes an experimental study of a fiber speckle sensor based on MMF. The sensor was evaluated by measuring different ratios of glycerin and water solution. The results show that the fiber speckle sensor provides a simple, efficient, and cost-effective approach to fiber sensing, which provides a reliable alternative to RI detection. This work has important implications for the expansion of fiber-optic sensing technology and has potential applications in areas such as oceanographic research. In general, the article is built very logically, consistently, and pleasantly read. As some of the issues that remained after reading the article, the following can be distinguished.
- The scale of Figure 6 may be increased for clearer reding the graphics. In present state the most volume of figure is empty.
A: To seem clearer for reding the image, the scale of Figure 6 in manuscript has been revised, and some detailed information has been added in the image, shown in Figure 9 in revised manuscript.
Figure 9. The experiment results of the FSS system temporal stability, the taper region of TMMF is immersed in air and water.
- The maximum for ZNCC in figures 5-7 must be limited by value 1.0.
A: As reviewer’s suggested, the maximum value ZNCC of Figures 6-10 in revised manuscript have been limited by 1.0.
Figure 6. The value of ZNCC with the variation of the different RI. A linear response curve displays a 6.41 RIU-1 sensitivity.
Figure 7. The linear response of ZNCC with RI under three offset conditions when SMF and MMF are spatially coupled. (a) FSS response with increased axial offset; (b) FSS response with increased lateral offset, as well as increased lateral and axial offset.
Figure 8. the response of the number of LP modes to RI FSS, inset is the simulation layout.
Figure 9. The experiment results of the FSS system temporal stability, the taper region of TMMF is immersed in air and water.
Figure 10. Multiple measurements of the state when sensing head is immersed in water.
- The text is needed in improving. Some sentences look like fragments, for example, “Because all excited high-order modes contribute to the speckle pattern.” “Or applying a lateral external force to an MMF resulting in a minor change in MMF diameter.”
A: We have removed the fragments sentences and checked the grammar mistakes in the revised manuscript. [The description is added to in revised manuscript line 183-189.]
These comments are not fundamental, and the article is recommended for publication in the journal after small corrections.

Reviewer 4 Report
Please check the attachment

Please check the attachment
Author Response
Reviewer 4
The paper Spatially Modulated Fiber Speckle for High-sensitivity Refractive Index Sensing describes a novel method of sensing small changes in refractive index, applicable to seawater salinity measurements. Sensing is realized by submersion of a tapered multimode fiber into the solution of interest and monitoring changes in the speckle patterns using a CMOS camera. The authors show that the more modes are contributing to the output speckle pattern, the more sensitive it is and the higher resolution can be achieved. The described method has a lot of potential, however I think the paper can’t be published in its current form and requires major revisions.
- The authors claim a very high resolution <10-4. However, how such high resolution relates to the sensor accuracy remains unclear. The sensor precision (stability of subsequent measurements) is addressed and is shown to be quite high, but it doesn't mean that the sensor is equally accurate, i.e. shows RI value close to actual. In fact, Figures 4 and 5 show that measured ZNCC values lie much further from the theoretical linear fit than one might expect knowing only about a sensor's high resolution. So these are the important questions that need to be answered to properly evaluate the sensor’s quality. Is the relationship between ZNCC and RI really linear?
A: We much appreciate the reviewers’ valuable questions. To investigate the relationship of ZNCC and RI variation, we have systematically reconducted experiments and done the related simulation. The related results have added in the revised manuscript as shown in Figure 5, 7 and 8. Both numerical stimulation and experiment exhibit linear and monotonic variations. The R2 of each linear fitting has been added in Figure 5 and 7, and all R2 are greater than 0.9.
Figure 5. The value of ZNCC with the variation of the different RI. A linear response curve displays a 6.41 RIU-1 sensitivity.
Figure 7. The linear response of ZNCC with RI under three offset conditions when SMF and MMF are spatially coupled. (a) FSS response with increased axial offset; (b) FSS response with increased lateral offset, as well as increased lateral and axial offset.
Figure 8. the response of the number of LP modes to RI FSS, inset is the simulation layout.
- What would it look like if one measured ZNCC for every 10-4 RIU?
A: Based on the linear fitting in simulation of 20 LP shown in Figure 8, every 10-4 changes in RI, 0.0015 changes in ZNCC. However, the variation values of experimental variables (vol % of glycerol and water) corresponding to every 10-4 changes are less than the range that can be simulated by the beyond experiment.
- If ZNCC(RI) is actually a line, then the points on figures 4 and 5 should be very close to that line. If it instead follows some smooth monotonous curve, then the sensor can be very accurate, but would require a rigorous calibration step to determine that curve. And if the curve is not monotonous and looks like a sum of a descending line and some high frequency noise, then the noise amplitude will determine the sensor accuracy, while the product of sensitivity and precision (that the authors use here to define resolution) does not really characterize the sensor. These are some other points that when properly addressed would improve the quality of the paper: As I understand, the authors use a speckle pattern measured at the lowest RI as reference, therefore all plots start with ZNCC=1. Is it correct, that for RI lower than the reference value, ZNCC would be lower than 1? In this case, ZNCC approach is intrinsically unable to distinguish between RI values lower and higher the reference value. How can it be solved? Should one store speckle patterns at many RI points to avoid this ambiguity? Can it improve accuracy?
A: Thanks for reviewer’ valuable comment. We discussed and added the limitations of ZNCC. The ZNCC can determine the difference in correlation between two images but limits in determining the specific values of the parameters that cause image correlation changes. [The description is added to in revised manuscript line 156-158.] If appropriate machine learning is introduced to ZNCC, and the initial reference value is determined independently by the computer and the prediction of the first target value is completed, then ZNCC will show great potential in improving its response range and accuracy.
- The authors found that exciting more modes, especially high-order modes improve the sensor’s sensitivity. While intuitively it aligns with theoretical predictions, this finding is shown only qualitatively. A proper quantitative theoretical analysis would be very interesting.
A: To reflect the sensitivity characteristics of FSS to a certain extent in our work, we do the numerical simulation on the relationship between the number of modes and sensitivity. As shown in Figure. 8, the simulation findings of the responses of various modes to RI change are exhibited. The simulated modeling is used here to qualitatively account for the impact of mode number on FSS sensitivity rather than to provide an accurate quantitative comparison with the experiment. The complex factors such as the temperature and noise will be considered in future.
Figure 8. the response of the number of LP modes to RI FSS, inset is the simulation layout.
- How temperature fits into the general picture described in the paper is unclear. Temperature changes could destroy speckle patterns and make it impossible to distinguish between RI changes and T changes. Again, it seems that a simple approach based on ZNCC measurement could be not enough in a practical setting. Measurements of multiple reference speckle patterns for different RI and T may in principle solve this problem though.
A: Thanks for reviewer’s comment. The relevant works have reported the use of FSS for temperature sensing including predication of air temperature and magnetic field sensing with temperature compensation [R4-R5]. After clarifying that FSS can be used to sense temperature and multimodality, our work has concentrated more attention on near physical work, namely the impact of modes on FSS sensitivity. To avoid the impact of temperature on the RI measurement results, we maintain the room temperature at 25 ℃. [The description is added to in revised manuscript line 210-212.]
In the final version of the manuscript, I would also recommend to address the following points:
- While in general the quality of English is all right and the authors’ message is clear, there are many places where the quality of English could be improved. There are some of them:
2-51: “power meter is measured”
2-56: “a new degree of free”
3-123: “mode corresponding to the real part of the effective refractive index smaller
than that of the fiber core refractive index”
4-154: “it is a statistical method” - could be rephrased
f4-156: “And it shows”
5-163: “are determined by” - should probably be replaced by “correspond to”
5-164: “speckle gram”
A: Thanks for the reviewer. We have rephrased the grammar mistakes in the whole manuscript.
- Quality of Figure 6 could be improved by changing to a line plot or making points smaller. The Y-axis scale could be changed to better show ZNCC instability and make comparison between two lines easier.
A: To seem clearer for reding the image, the scale of Figure 6 has been revised, and some detailed information has been added in the image, shown in Figure 9 in revised manuscript.
Figure 9. The experiment results of the FSS system temporal stability, the taper region of TMMF is immersed in air and water.
- Mathematical formulas should be checked for typos and made clearer:
For example k - means different things in 3-97 and 3-98.
It should be φm−φn in 3-98.
“Approximately equal” sign should be used in 3-108.
It is not clear what the summation means in 4-161.
A: We have revised the mathematical typos in the whole manuscript.
[R4] Smith, D. L., Nguyen, L. V., Ottaway, D. J., Cabral, T. D., Fujiwara, E., Cordeiro, C. M., & Warren-Smith, S. C. (2022). Machine learning for sensing with a multimode exposed core fiber specklegram sensor. Optics express, 30(7), 10443-10455.
[R5] Zhu, R. Z., Wan, S. J., Xiong, Y. F., Feng, H. G., Chen, Y., Lu, Y. Q., & Xu, F. (2021). Magnetic field sensing based on multimode fiber specklegrams. Journal of Lightwave Technology, 39(11), 3614-3619.

Round 2
Reviewer 1 Report
I would like to thank authors for taking extensive effort in not only addressing reviewers concerns and questions but also revising the experiments to make the study more systematic and impactful. The revision has clearly improved the understanding of the sensing and interrogation method and will be helpful to the readers. If the corrections for temperature and vibration/noise-induced effects on speckle patterns can be made to increase its robustness, the ZNCC interrogation can provide cheaper alternative options to spectrometers-based interrogation for RI sensing in fiber-optics sensors.